# Immune-Related Mutational Landscape and Gene Signatures: Prognostic Value and Therapeutic Impact for Head and Neck Cancer

**DOI:** 10.3390/cancers13051162

**Published:** 2021-03-08

**Authors:** Bohai Feng, Jochen Hess

**Affiliations:** 1Department of Otorhinolaryngology, Head and Neck Surgery, Heidelberg University Hospital, 69120 Heidelberg, Germany; bohaifeng91@gmail.com; 2Department of Otorhinolaryngology, Second Affiliated Hospital of Zhejiang University School of Medicine, Hangzhou 310009, China; 3Research Group Molecular Mechanisms of Head and Neck Tumors, German Cancer Research Center (DKFZ), 69120 Heidelberg, Germany

**Keywords:** CNA, EGFR, HNSCC, ICI, IFN-γ, immune-related gene signature, PD-1, PD-L1, TIME, TMB

## Abstract

**Simple Summary:**

Immunotherapy has emerged as a standard-of-care for most human malignancies, including head and neck cancer, but only a limited number of patients exhibit a durable clinical benefit. An urgent medical need is the establishment of accurate response predictors, which is handicapped by the growing body of molecular, cellular and clinical variables that modify the complex nature of an effective anti-tumor immune response. This review summarizes more recent efforts to elucidate immune-related mutational landscapes and gene expression signatures by integrative analysis of multi-omics data, and highlights their potential therapeutic impact for head and neck cancer. A better knowledge of the underlying principles and relevant interactions could pave the way for rational therapeutic combinations to improve the efficacy of immunotherapy, in particular for those cancer patients at a higher risk for treatment failure.

**Abstract:**

Immunotherapy by immune checkpoint inhibition has become a main pillar in the armamentarium to treat head and neck cancer and is based on the premise that the host immune system can be reactivated to successfully eliminate cancer cells. However, the response rate remains low and only a small subset of head and neck cancer patients achieves a durable clinical benefit. The availability of multi-omics data and emerging computational technologies facilitate not only a deeper understanding of the cellular composition in the tumor immune microenvironment but also enables the study of molecular principles in the complex regulation of immune surveillance versus tolerance. These knowledges will pave the way to apply immunotherapy more precisely and effectively. This review aims to provide a holistic view on how the immune landscape dictates the tumor fate and vice versa, and how integrative analysis of multi-omics data contribute to our current knowledge on the accuracy of predictive biomarkers and on a broad range of factors influencing the response to immunotherapy in head and neck cancer.

## 1. Introduction

Head and neck cancers (HNCs) are among the most frequent and destructive human cancers worldwide, causing considerable morbidity and mortality [1,2]. Head and neck squamous cell carcinomas (HNSCCs) account for the majority of HNC, are unexpectedly heterogeneous in nature, and tobacco use, extensive alcohol consumption and infection with high-risk human papillomavirus (HPV), in particular HPV16, are the main etiological risk factors [3,4,5]. As for many other solid cancers, HNSCC pathogenesis resembles a tightly orchestrated balance between immune effector response and tolerance, and cancer cells evade the host immune surveillance by a broad range of cellular and molecular mechanisms [6,7,8]. Hence, the tumor immune microenvironment (TIME) of individual HNSCCs is rather heterogeneous and characterized by a broad spectrum of qualitative and quantitative differences in immune cell populations [9].

Despite continuous improvement in conventional treatments, consisting of surgery, radio- and chemotherapy, a substantial proportion of HNSCC patients suffer from locoregional relapse or distant metastasis [3,10]. For patients with recurrent or metastatic disease (R/M-HNSCC) the armamentarium of systemic anti-cancer modalities and innovative local approaches continues to grow, but the overall survival remains dismal and is still unsatisfactory [11,12]. In the past decade, immunotherapy based on immune checkpoint inhibition (ICI) has become an essential pillar for cancer treatment and now represents the standard of care for most human cancers, including HNSCC [3,13]. Activation of immune checkpoint cascades such as those controlled by cytotoxic T lymphcytes associated protein 4 (CTLA-4) or programmed cell death protein 1 (PD-1) and its ligand (PD-L1) results in inactivation of tumor-specific T cells and immune evasion. The underlying concept of ICI is that treatment with anti-PD-(L)1 or anti-CTLA-4 antibodies reinvigorates cytotoxic immune cells to target cancer cells [14,15,16].In randomized phase III trials two monoclonal antibodies (nivolumab and pembrolizumab), targeting PD-1 demonstrated longer overall survival in comparison with standard chemotherapy in pretreated R/M-HNSCC [17,18].Furthermore, pembrolizumab demonstrated superiority relative to standard first-line cytotoxic treatment for R/M-HNSCC, either as single-agent therapy or in combination with chemotherapy for tumors with PD-L1 expression [19].However, a major limitation is the low overall response rate of ICI therapy and many patients have experienced minimal or no clinical benefit [13,20]. Due to the relatively poor response rate, potential risk for hyper-progressive disease, and high degree of immune-related adverse events (irAEs), an urgent medical demand exists for reliable cellular or molecular biomarkers (Figure 1) to support treatment-decision making and a better stratification of cancer patients at higher risk for intrinsic or acquired treatment failure, who might benefit from new strategies of combination therapies [21,22].

## 2. Response Evaluation and Biomarker Development for ICI

The therapeutic activity of ICI is the consequence of a complex interplay between cancer cell intrinsic traits, the tumor microenvironment (TME) and the host immune system (Figure 2) [23]. Though multiple factors affect ICI effectiveness, only few biomarkers have been established for response evaluation and risk assessment of patient prognosis [24]. These emerging biomarkers can be categorized in molecular features, which are either related to the tumor neoepitope burden, including microsatellite instability (MSI) or high tumor mutational burden (TMB), or those resembling a T cell–inflamed TME [24]. AT cell–inflamed TME often exhibits high PD-L1 protein levels on tumor and immune cells as a consequence of local T cell–derived interferon-γ (IFN-γ), and prominent expression of gene signatures of activated T cells [25].

Detection of PD-L1 expressing tumor cells and to a lesser extent immune cells by immunohistochemical staining serves as a valid biomarker in some but not all cancer types [24]. For R/M-HNSCC patients, the KEYNOTE-040 trial demonstrated that the benefit of pembrolizumab was greater in patients with PD-L1 expressing tumors [18]. However, a significant benefit was also observed for patients with PD-L1-negative tumors (KEYNOTE-012, KEYNOTE-055) [26,27]. In an exploratory biomarker analyses the survival benefit from nivolumab (CheckMate-141) was seen regardless of the tumor PD-L1 status, but the magnitude of benefit was greater when tumor PD-L1 expression was ≥1% [17]. These data strongly suggest that other factors in addition to PD-L1 expression might serve as important predictive biomarkers [28].

One emerging predictive biomarker for clinical response to ICI therapy is the TMB (the total number of mutations per coding area of a tumor genome), which has been evaluated in numerous clinical trials across many human cancers, including HNSCC [29,30,31]. The predictive value of TMB is further supported by the clinical activity of immunotherapy in colon cancers with mismatch repair deficiency, a tumor subtype with a high TMB, as compared to mismatch repair proficiency counterparts with significantly lower TMB and a poor response to ICI [32]. A high TMB is strongly associated with improved patient survival and its predictive value is often superior to the assessment of PD-L1 expression [32]. In particular, tumors with a higher non-synonymous mutation burden share a consistently improved responses to ICI therapy as such mutations are a potential source of neoantigen epitopes. The non-synonymous TMB is strongly correlated with the median global TMB, but varies among cancer types and individuals [30]. In a recent study with 1184 HNSCC, the median number of somatic mutations was 5.0 per mega bytes (Mb), and 10.1% of patients were classified as high TMB (>20 mutation/Mb) [33]. However, these numbers are context dependent and vary among studies [34,35]. In addition to a higher TMB, somatic frameshift events in tumor suppressor genes predict ICI response of HPV16-negative HNSCC [36]. It is worth noting that somatic mutation rates across genomes or exomes of HPV-positive oral and oropharyngeal cancers did not differ significantly from HPV-negative counterparts [34]. Accordingly, the impact of the HPV16 status on relative efficacy of ICI treatment is still under debate, though initial data indicated that the magnitude of benefit with nivolumab or pembrolizumab might be greater for p16-positive oropharyngeal cancers [17,26,27] (see Section 5 for a more detailed discussion). It is worth noting that TMB does not always correlate with ICI responsiveness and its applicability should be considered with caution due to important limitations as a predictive biomarker, especially when used in isolation. Major challenges for TMB utility and its limitations have been reported and critically discussed in excellent recent studies and review articles [29,37,38,39,40]. A composite predictor that also includes other critical variables, such as PD-L1 IHC, immune-related mutational and epigenetic landscapes as well as gene expression signatures, MHC and T cell receptor repertoire, clonality of neoantigens and tumor heterogeneity, is urgently needed [33,38].

## 3. Immune-Related Mutational and Epigenetic Landscapes

TMB and high microsatellite instability are indirect measures of tumor antigenicity generated by somatic mutations (Figure 3). Somatic mutations in tumor DNA can induce neoantigens production, which can be targeted and recognized by the immune system, particularly after treatment with agents that activate T cells [29]. Those somatic mutations are transcribed and translated, and neoantigen-containing peptides are processed by the antigen-processing machinery and are loaded onto MHC molecules to be presented on the cell surface. However, not all somatic mutations produce peptides which are appropriately processed and loaded onto MHC complexes, and even fewer can be recognized by T cells [7,41,42]. Recent studies have revealed mutational signatures underlying the evolution of cancer and highlighted a strong association of HPV with APOBEC mutational signatures, suggesting impaired antiviral defense as a driving force in distinct cancers, including HNSCC [41,42,43]. Almost all HPV-positive and many HPV-negative HNSCCs share a large fraction of somatic mutations attributable to members of the apolipoprotein B mRNA editing enzyme catalytic subunit-like protein 3 (APOBEC3) family of single-stranded DNA cytosine deaminases [44,45]. Utilizing whole-exome and RNA-seq datasets from The Cancer Genome Atlas (TCGA), Faden et al. [46] observed the highest IFN-γ levels for HNSCC across cancer types with high APOBEC-related mutational burden. Most prominent IFN-γ scores in HNSCC were present in HPV-related tumors and tumor-specific neoantigens were significantly correlated with mutational burden attributed to APOBEC [46]. In another study, a subgroup of APOBEC-enriched, HPV-negative HNSCC with a distinct immunogenic phenotype was identified, which was characterized by higher T-cell inflammation, prominent immune checkpoint expression and enrichment of mutations in immune-evasion pathways [47].

In contrast to TMB, a high-level of aneuploidy also known as somatic copy number alterations (SCNAs) correlates with markers of immune evasion and with reduced response to immunotherapy (Figure 3). A higher burden of copy number loss in non-responders to CTLA-4 and PD-1 blockade was identified in a cohort of melanoma patients, which was associated with decreased expression of genes in immune-related pathways [48]. Davoli et al. [49] investigated 12 cancer types from TCGA to demonstrate that most highly aneuploid tumors exhibit reduced expression of markers for infiltrating immune cells, especially CD8-positive T cells and NK cells, indicating that aneuploidy restricts cytotoxic immune response during tumorigenesis. Again, tumor aneuploidy inversely correlates with patient survival in two clinical trials of ICI therapy for metastatic melanoma [49]. The effect on treatment response for TMB and aneuploidy was non-redundant, suggesting that both alterations reflect different aspects in the balance of immune surveillance versus tolerance and the potential utility of a combinatorial biomarker to optimize patient care with ICI therapy [48]. In line with this assumption, the combination of TMB and CNA for prognostic risk assessment and prediction of heterogeneous clinical responses to ICI treatment was confirmed for multiple cancers [50,51]. A high level of aneuploidy was also found for an immune cold subtype with least amount of tumor infiltrating lymphocytes (TILs) based on a pan-SCC cohort of TCGA [52]. For primary HNSCC, an inverse correlation between copy number alteration and measures of immune infiltration was evident [53,54], and a lower cytotoxic immune phenotype exhibited a characteristic pattern of copy number loss affecting chemokine signaling and immune effector response [55].

Profound global loss of DNA methylation is a hallmark of many cancers, and global demethylation in cancer promotes chromosomal instability [56,57], particularly involving large-scale alterations leading to aneuploidy [58]. Cancers commonly hijack various epigenetic mechanisms to escape immune restriction, but the impact of DNA methylation on immune evasion and in the context of cancer immunotherapy has been addressed only recently (Figure 3) [59,60,61]. In a pan-cancer analyses of TCGA data, Jung et al. [62] found that genomic hypomethylation correlated not only with aneuploidy but also immune escape signatures independently of the mutational burden, and was associated with increased immunotherapeutic resistance. Moreover, inactivating mutations in the nuclear receptor binding SET domain protein 1 (NSD1) a histone methyltransferase define an intrinsic subtype of HNSCC that features pronounced DNA hypomethylation [35,63,64] and displays an immune cold phenotype characterized by low levels of TILs and low expression of a CD8-positive T cell inflamed gene signature [65,66]. These data indicate NSD1 as a tumor cell-intrinsic driver of an immune cold phenotype by causing epigenetic deregulation with potential implications for immunotherapy (Figure 3).

Reactivation of transposable elements (TEs) including endogenous retroviral (ERV) transcripts is another consequence of profound global loss of DNA methylation in cancer. It results in a state of viral mimicry in which treated cancer cells mount an immune response by turning on viral defense genes and potentially expressing neoantigens [67]. In a pan-cancer analysis with TCGA cohorts, expression of 262 TE subfamilies appear to result from a proximal loss of DNA methylation [68]. TE overexpression in tumor samples with respect to matched normal controls is most prominent in stomach, bladder, and liver cancer as well as HNSCC. At the global level, this overexpression in HNSCC is associated with loss of DNA methylation, particularly at proximal CpG sites, suggesting targeted loss of DNA methylation near TE sites as a major mode of regulation [68]. For HNSCC, tumors with a high ERV expression pattern share prominent immune checkpoint pathway activation and increased immune infiltration with a higher CD8-positive T cell fraction as compared with ERV low expressing counterparts [69]. These data together with recent preclinical studies provide a strong rational for combining epigenetic targeting and immune checkpoint blockade in HNSCC to enhance treatment efficacy. Due to pleiotropic effects on multiple targets, which could limit the risk for treatment resistance, inhibition of epigenetic modifications emerges as promising strategy in combination with ICI [70]. Indeed, increased TE expression and de novo presentation of TE-derived peptides on MHC class I molecules were found upon treatment of cancer cells with a demethylation agent, indicating that therapeutic reactivation of tumor-specific TEs may synergize with immunotherapy. In line with this assumption, the phase IbNIBIT-M4 trial reported that treatment of patients with advanced melanoma using the next-generation DNA hypomethylating agent guadecitabine combined with ipilimumab is safe and tolerable, and shows promising immunomodulatory and antitumor activity [71].However, in an open-label phase II multi-cohort study administration of the oral DNA hypomethylating agent CC-486 combined with durvalumab did not demonstrate robust pharmacodynamic or clinical activity in selected immunologically cold solid tumors consisting of PD-(L)1 inhibitor naïve patients with either advanced microsatellite stable colorectal cancer, platinum resistant ovarian cancer, or estrogen receptor positive, HER2 negative breast cancer [72].

Enhancer of zeste homolog 2 (EZH2), a methyltransferase subunit of the polycomb repressive complex 2 (PRC2) that catalyzes histone H3 methylation on lysine 27 (H3K27), represents another epigenetic target to circumvent ICI resistance in HNSCC [73,74]. EZH2 expression was negatively correlated with components of the antigen-processing machinery pathway in TCGA-HNSC and genetic ablation or pharmacological inhibition of EZH2 resulted in a significant increase of MHC class I expression on HNSCC cells, antigen-specific CD8-positive T cell proliferation, IFN-γ production, and tumor cell cytotoxicity. In a preclinical mouse model, the combination of an EZH2 inhibitor (GSK126) and anti-PD1 antibodies suppressed tumor growth of anti-PD-1-resistant HNSCC [74]. The association of chromatin modification with CD8-positive T cell exclusion in HPV-negative HNSCC was further supported by a study of Vougiouklakis et al. [75]. They identified a couple of protein methyltransferases (PMTs) and demethylases (PDMTs) with inverse expression pattern compared to components of the antigen presentation machinery, CD8-positive T cells and immune-active chemokines. Finally, a phase II trial of pembrolizumab and vorinostat, a pan-HDAC (histone deacetylase) inhibitor, with progressing and incurable head and neck cancers demonstrated activity in R/M-HNSCC, but fewer responses in salivary gland cancer [76]. However, toxicities were higher than reported with pembrolizumab alone and no complete responder was observed.

## 4. Immune-Related Gene Signatures

### 4.1. Pan-Cancer Studies

The TIME plays not only a critical role in neoplastic transformation, malignant progression and metastasis, but also is a key determinant of therapy response and prognosis of most cancers, including HNSCC [77,78]. However, the underlying molecular principles driving the establishment and maintenance of the TIME are complex (Figure 2) and their elucidation could help in finding new ways to treat cancer or to improve the effectiveness of immunotherapy [28,79,80]. Though traditionally the spectrum of immune cell infiltrates has been assessed using antibody staining and microscopic techniques or FACS, recent advances in genomic technologies and bioinformatics approaches in combination with the availability of large multi-omics data have facilitated the systemic dissection of molecular principles, how the mutational landscape or the epigenome of a tumor shapes the host immune system [7,81,82] or vice versa (Table 1, Figure 3). Most popular computational tools to quantify immune cells from expression data of bulk tumor tissue make use of selected gene sets coupled with gene set enrichment analysis (GSEA) or similar scoring approaches, or leveraging on a signature matrix describing the cell type-specific expression profiles combined with deconvolution algorithms [81]. Using RNA-seq data from 18 TCGA cancer cohorts, Rooney et al. devised a simple and quantitative measure of immune cytolytic activity based on transcript levels of granzyme A (GZMA) and perforin (PRF1), which are upregulated upon cytotoxic T cell activation and during clinical responses to either anti-CTLA-4 or anti-PD-L1 immunotherapies [83]. A higher cytolytic activity was associated with somatic mutations in genes involved in antigen-presentation (e.g., HLA, B2M) or extrinsic apoptosis (e.g., CASP8), indicating a mode of positive selection rendering affected tumors resistant to immune surveillance [50,83]. A close association between CASP8 somatic mutations with either prominent immune cell infiltrates, cytotoxic immune or T cell inflamed phenotypes was confirmed in TCGA-HNSC as well as independent HNSCC cohorts [55,66,84]. HLA mutations occur in roughly 5% of HNSCCs, and loss of function or expression are associated with deregulation of innate antiviral and adaptive antitumor immunity [35,85,86,87]. It is also worth noting that the composition of HLA class I genotypes influences the response to immunotherapy. In a cohort of 1535 advanced cancer patients, maximal heterozygosity at HLA class I loci improved overall survival after ICI therapy, while somatic loss of heterozygosity at HLA class I loci was associated with poor outcome [88].

More recently, the immune landscape of cancer has been explored by two groups based on an extensive immunogenomic analysis of around 10,000 tumors comprising diverse cancer types utilizing data compiled by TCGA. Across cancer types, Thorsson et al. [91] identified six molecular immune subtypes: wound healing (C1), IFN-γ dominant(C2), inflammatory (C3), lymphocyte depleted (C4), immunologically quiet (C5), and TGF-β dominant (C6). These subtypes were characterized by differences in macrophage or lymphocyte signatures, Th1:Th2 cell ratio, extent of intra-tumoral heterogeneity, aneuploidy, extent of neoantigen load, overall cell proliferation, expression of immunomodulatory genes, and prognosis. It is worth noting that most tumors of the TCGA-HNSC cohort were categorized in C1 with elevated expression of angiogenic genes, a high proliferation rate, and a Th2 cell bias to the adaptive immune infiltrate or C2 with the highest M1/M2 macrophage polarization, a strong CD8 signal and, prominent TCR diversity. C2 also showed a high proliferation rate, which may override an evolving type I immune response in cancers, including HNSCC [91]. Tamborero et al. [92] also identified six immunophenotypes across cancer types and characterized genomic and transcriptomic traits associated to individual immunophenotypes. In this study, a substantial fraction of HNSCC were detected in all six subtypes, but a higher cytotoxic immunophenotype was not associated with improved survival. A lesion from both studies is that multiple control modalities of molecular networks affect tumor-immune interactions and might influence response to ICI therapy.

Immune-related gene expression profiles (GEPs) are tissue-agnostic measures of distinct aspects of tumor immunobiology and can predict either alone or in combination with TMB or PD-L1 expression the response to ICI therapy across multiple tumor types (Figure 3). Ayers et al. [90] analyzed several GEPs using RNA from baseline tumor samples of pembrolizumab-treated patients to identify immune-related signatures correlating with clinical benefit using a learn-and-confirm paradigm. A pan-tumor T cell-inflamed GEP has been established and was independently confirmed and compared with that of PD-L1 immunohistochemistry in HNSCC patients. The T cell-inflamed GEP contains IFN-γ-response genes related to antigen presentation, chemokine expression, cytotoxic activity, and adaptive immune resistance, and has been developed into a clinical-grade assay [90]. Meanwhile, the predictive value of the T cell-inflamed GEP in combination with TMB or other inflammatory biomarkers (e.g., PD-L1 expression) has been confirmed for clinical response to ICI therapy across a broad spectrum of cancers, including HNSCC [94,95]. It is worth noting that TMB was neither significantly associated with the T cell-inflamed GEP nor with PD-L1 expression in the HNSCC cohort. This is in accordance with other reports supporting that TMB does not significantly correlate with cellular or molecular immune phenotypes in HNSCC, indicating that the presence of neoantigens might be a necessary but not a sufficient factor to mount an effective anti-tumor immunity in HNSCC patients [54,55,66].In contrast, the T cell-inflamed GEP was significantly correlated with PD-L1 in HNSCC, similar to the pan-cancer cohort, consistent with the known regulation of PD-L1 gene expression by IFN-γ from activated T cells [95].

Finally, Jiang et al. [93] developed a computational tool, called TIDE (Tumor Immune Dysfunction and Exclusion) to model tumor immune evasion based on integrative analysis of gene expression signatures resembling T cell dysfunction in tumors with high infiltration of cytotoxic T lymphocytes (CTL) or the prevention of T cell infiltration in tumor tissue (Figure 3). Though TIDE was trained from treatment-naive tumor data, its predictive accuracy for melanoma patients treated with ICI was superior as compared to other biomarkers, such as PD-L1 or TMB. In addition, potential regulators of ICI resistance have been elucidated and one candidate gene (SERPINB9) was experimentally confirmed in pre-clinical cell culture and mouse models [93]. However, the potential of TIDE as a reliable surrogate biomarker to predict ICI response for other cancers, including HNSCC remains to be demonstrated.

### 4.2. HNSCC Studies

Several groups utilized complex bioinformatics and computational algorithms to analyze transcriptomic data from TCGA-HNSC, often in combination with independent validation HNSCC cohorts, to identify molecular immune tumor subtypes based on altered immune-related gene expression profiles (Table 1, Figure 3). Applying unsupervised clustering of gene expression data, Keck et al. [89] identified an immune mesenchymal subtype in HPV-positive and HPV-negative tumors that was associated with increased expression of immune markers and higher levels of CD8-positive lymphocytes. Mandal et al. [54] revealed a broad and context dependent diversity in levels of immune infiltration and activation across tumors, but also that HNSCCs are among the most highly immune-infiltrated cancer types. At the same time, a substantial amount of HNSCC, particularly HPV-positive tumors, shared high levels of immunoregulatory features, such as prominent Treg infiltration, indicating that these tumors are poised to respond to immunotherapeutic modalities that relieve inhibitory pathways [54].

Messinaet al. [96] implemented a 12-chemokine gene signature, which was related with induced CD8-positive T cell infiltration and overall survival in melanoma metastases, to analyze multi-omics data of two cohorts with primary HNSCC, TCGA-HNSC and the Chicago Head and Neck Genomics (CHGC) cohort [66]. While HNSCC with a low CD8-positive T cell inflamed phenotype were enriched for β-catenin and Hedgehog pathways, NSD1 mutations and EGFR amplifications, a high CD8-positive T cell inflamed phenotype was associated with MAPK/ERK and JAK/STAT pathways, CASP8 mutations and CD274 amplifications [66].

Chen et al. [53] separated gene expression patterns from tumor, stromal, and immune cell genes using a non-negative matrix factorization algorithm and correlated the expression patterns with a set of immune-related gene signatures, potential immune biomarkers, and clinicopathological features. Approximately 40% of tumors in the TCGA-HNSC cohort shared an enriched inflammatory response, enhanced cytolytic activity, and active IFN-γ signaling. This immune class could be divided into two distinct microenvironment-based subtypes, characterized by markers of active or exhausted immune response. The robustness of these molecular immune subgroups was verified in independent HNSCC validation cohorts, and the active immune subtype showed potential response to PD-1 blockade in a melanoma cohort [53].

However, most computational algorithms that have been conducted to stratify molecular immune subgroups focused on characteristic features of cytotoxic T cells, which could not fully reflect the complexity of the TIME involved in immune evasion or the response to immunotherapy (Figure 2). Hence, we trained a novel molecular classifier based on those immune cell subsets strongly associated with both PD-L1 and IFN-γ expression in TCGA-HNSC as well as independent HNSCC cohorts [55]. This strategy was in line with the assumption that high IFN-γ levels as an important regulator of PD-L1 expression accompanied by elevated levels of TILs may be the key to identify immunologically hot tumors, which are more likely to respond to ICI therapy [25,97]. We identified subgroups with hot and cold immune phenotypes based on the relative abundance of five immune cell types (CD8 T cells, activated CD4 memory T cells, activated NK cells, M1 macrophages, and M2 macrophages), and integrative analysis of multi-omics data elucidated the epidermal growth factor receptor (EGFR) and the prostaglandin-endoperoxide synthase 2 (PTGS2) as key nodes in a gene regulatory network related to the immune cold phenotype [55]. An association between EGFR activity and an immune cold phenotype has been previously reported in multiple cancers, including HNSCC [50,66,98,99], and clinical trials reported lower benefit from ICI therapy for non-small cell lung cancer with EGFR mutations [100,101]. Compelling experimental and clinical evidence also demonstrated that EGFR signaling actively regulates the tumor immune microenvironment and that EGFR inhibition prompts not only an increase in TILs and expression of immune checkpoints, but also serves as a promising immunotherapy sensitizer [102,103,104,105]. An active interaction between COX2, encoded by PTGS2 and the EGFR pathway is well-established in carcinogenesis [106,107], and similar to EGFR, COX2 activity is related to impaired immune surveillance and cancer immune evasion by triggering immunosuppressive properties of diverse cells in the TIME [108,109,110].

While the presence of abundant tumor infiltrating lymphocytes is generally associated with improved prognosis (Figure 3), differences have been reported according to anatomic subsite, tumor compartment and depending on the HPV status [28]. Numerous groups addressed the prognostic value of immune-related gene signatures in primary HNSCC and provided growing evidence for a pivotal correlation between molecular immune subgroups and prognostic risk assessment for HNSCC patients [111,112,113,114,115,116,117,118,119,120]. Though most studies made use of similar transcriptome and clinical data, in particular from TCGA-HNSC the diversity of clinically-relevant immune-related gene signatures is quite high with only minor or no overlap in selected candidate genes. However, presented data support the assumption that HNSCC utilize multiple immune escape mechanisms and thus underline the importance of multitargeted schedules to improve the potential of immunotherapy in future clinical trials.

## 5. Immune Landscape of HPV-Positive Versus HPV-Negative HNSCC

The duality of carcinogen- versus virus-induced cancers is an important aspect of HNSCC and presents a unique opportunity to assess differences in the immune landscape of two distinct cancer etiologies that occur in a similar anatomical region. Numerous studies demonstrated that HPV-positive OPSCCs exhibit an immunologically distinct subgroup within HNSCC, and it has been speculated that the favorable survival as compared to their HPV-negative counterparts reflects at least in part a more active immune contexture, which is conditioned by the virus [121]. A more active immune contexture in combination with the presence of viral antigens has raised hope that the response rate of HPV-positive tumors to ICI therapy with an anti-PD-(L)1 antibody is comparatively high. Indeed, patients enrolled in KEYNOTE-012 with HPV-positive tumors had a higher overall response rate (ORR) as compared to HPV-negative counterparts, which was irrespective of the PD-L1 status [26]. A similar benefit in ORR was evident in the HAWK trial, which included immunotherapy-naïve patients with R/M-HNSCC and high tumor PD-L1 expression [122]. However, ORR among patients with HPV-positive versus HPV-negative tumors was similar in KEYNOTE-055 [27] and several clinical trials, including KEYNOTE-040 [18], KEYNOTE-048 [19], CheckMate141 [17,123], and the EAGLE trial [124], have reported benefit from ICI therapy regardless of the HPV status. In summary, these studies do not provide strong evidence that patients with HPV-positive tumors (as determined for most cases by p16 IHC staining) experience a distinct benefit by currently available ICI treatment. Consequently, the consensus recommendation of the Society for Immunotherapy of Cancer from the year 2019 states that the HPV status should not affect selection of patients with platinum-refractory R/M HNSCC for ICI therapy [13]. This situation might change with a better understanding of cellular and molecular features shaping the TIME of HPV-positive OPSCC and the implementation of combinatorial treatment strategies.

Main features of HPV-related differences that shape the TIME and might influence the response to immunotherapy have been discussed in several review articles [28,125,126,127,128]. An emerging picture from most studies indicates that comparing the immune profiles in HPV-positive versus HPV-negative HNSCC could pave the way in prioritizing which cell types and molecules to target for the development of novel concepts taking into account the biological context [129]. More recently, Cillo et al. [9] conducted an in-depth analysis of all CD45-positive immune cells in the TIME of HNSCC patients with either HPV-negative or HPV-positive tumors. They utilized single-cell RNA sequencing (scRNA-seq) analysis complemented by multispectral immunofluorescence to provide insight into distinct immune lineages, their transcriptional states and differentiation trajectories, and to pursue their spatial localization patterns and cellular cross-talk with potential relevance to tumor progression. One highlight of this study is that immune cells display a broad spectrum of transcriptional signatures, confirming a rather divergent pattern of CD4 T-helper cells, B cells and myeloid cells, while CD8-positive T cells and CD4-regulatory T cells are relatively similar among HPV-negative and HPV-positive HNSCC. It is also worth noting that this scRNA-seq approach did not recover immune cells consistent with a myeloid-derived suppressor cell (MDSC) phenotype in HNSCC with or without HPV. A higher frequency of intra-tumoral B cells in HPV-positive HNSCC is in line with previous reports [130,131,132], and the multispectral imaging analysis uncovered ternary lymphoid structures (TLS)in regions with high numbers of B cells [9]. The presence of TLS has been linked with improved survival across many cancer types, including HNSCC [133], and several studies provide compelling clinical evidence for the potential role of B cells and TLS in the response to ICI treatment, with implications for the development of new biomarkers and therapeutic targets [134,135].

HPV-related differences in the TIME of HNSCC are potentially due to the presence of viral antigens throughout carcinogenesis, leading to activation of innate immune responses early on and enhanced T and B cell-adaptive immune responses [9]. In support of this assumption, several recent studies have demonstrated intra-tumoral and virus-specific T cell or B cell responses, including HPV-specific antibodies, as a common feature of most HPV-positive OPSCC [136,137,138]. Of note, adaptive and humoral immune responses are not limited to viral E6 and E7 oncoproteins, but are triggered against a broad array of HPV-specific antigens [137,138]. Furthermore, the treatment status has the most significant impact on virus-related T cell immunity as the breadth and overall strength of HPV-specific T cell responses were significantly higher before the commencement of curative treatment than after therapy [137]. This might be explained by immunosuppressive effects of chemoradiotherapy, which impairs HPV-specific T cell immunity, or the resolution of an active disease following elimination of the potential source of antigens required to maintain HPV-specific T cell immunity. In summary, these data indicate that viral antigens trigger a tumor-specific adaptive and humoral immune response that shapes a favorable immune contexture in the TIME of HPV-positive primary OPSCC for a more effective response to therapy in the curative setting.

## 6. Immune Landscape and Field Cancerization

Oral leukoplakia and erythroplakia share multiple, clonally unrelated premalignant cells as a consequence of carcinogen-induced field cancerization, which are often clinically silent and could appear distant from an oral premalignant lesion (OPL) [139]. The concept of field cancerization was introduced by Slaughter et al. [140] and describes the replacement of normal cells by a cancer-primed cell population with minor or no morphological changes. These atypical epithelial cells are separated from the subepithelial stroma by the basement membrane. Alterations in the composition of the basement membrane as well as dermal extracellular matrix are early events in progression of oral premalignant lesions [141,142]. Partial destruction of the basement membrane integrity is not only augmented by extensive infiltration of lymphocytes and inflammatory cells, but also enables more effective immune cell infiltration from the subepithelial stroma into the stratified mucosal epithelium with atypical preneoplastic keratinocytes.The particular significance of field cancerization in tobacco-related HNSCC is the frequent occurrence of local recurrence and second primary tumor from those precancerous tissue after local treatment [139,143]. Meanwhile, field cancerization has been recognized as an underlying principle in the development of numerous human cancers, and is both enabled by and causes changes to the tissue microenvironment [144,145]. Fields of cancer-primed cell population are best characterized by their phenotypic traits, including properties such as an increased growth rate, decreased death rate, inflammation or increased immune evasion [145]. The latter property is at least in part manifested by altered composition of immune cell populations and expression of immune checkpoint proteins in OPLs [146]. As an example, PD-L1 is not only upregulation in OPLs, but is also associated with inferior oral cancer–free survival, suggesting a PD-L1–mediated mechanism of immune evasion at the preinvasive stage [146]. This assumption is supported by a recent meta-analysis, which reported a prevalence of PD-L1 expression in almost 50% of OPLs [147]. However, the range of PD-L1 expression among different studies is heterogenous and most failed to assess the impact of PD-L1 on subsequent oral cancer development. More recently, modulation of other immune checkpoints, including CD40/CD40LG and CTLA-4 pathways has been reported upon malignant transformation of human oral epithelium, suggesting multiple checkpoint pathways play a role in OSCC immune evasion [148].

The potential role of the PD-(L)1 immune checkpoint for malignant transformation of HNSCC was addressed by independent groups utilizing the 4-nitroquinoline-1-oxide (4-NQO) mouse model of carcinogen-induced oral carcinogenesis [149,150]. In this model, PD-1 blockade showed an encouraging efficacy in reducing the incidence of dysplastic lesions and prevented malignant progression to OSCC. Low-grade dysplastic lesions responded to PD-1 blockade with accumulation of activated T cells, which was accompanied by the induction of epithelial cell apoptosis in oral lesions [149,150]. In a follow-up study, Monteiro de Oliveira Novaes et al. [148]. demonstrated that treatment of mice bearing carcinogen-induced OPLs with a CD40 agonist decreased the incidence of invasive cancers more potently than any of the other immunotherapies evaluated, including treatment with an anti-PD-1 antibody as single agent or in combined with an anti-CTLA-4 antibody. Both treatments, PD-(L)1 pathway blockade or activation of the CD40 pathway were able to prevent OPL progression into invasive OSCC, but exhibited distinct patterns of immune modulation in the preclinical mouse model [148]. In summary, these results support the potential clinical benefit of immune checkpoint modulation to prevent OSCC development or to reduce the high risk of local recurrence and second primary tumor in tobacco-related HNSCC. Of note, immunoprevention has the advantage of targeting precancerous lesions regardless of the dominant dysregulated molecular pathways driving carcinogenesis.

## 7. Conclusions and Perspectives

The integrative analysis of multi-omics data is a powerful tool to elucidate immune-related cellular and molecular profiles with a strong impact on cancer immune escape and to establish predictive biomarkers for ICI response or resistance (Figure 1). However, a common limitation of most studies is the use of publicly available data sets from treatment naïve cancers to train respective models, even though clinical outcome is often unrelated to immunotherapy. An urgent demand for future studies is the availability of molecular profiling and clinical data from larger cohorts of HNSCC patients, which are gathered prior and ideally during ICI therapy. This will enable the elucidation of the complex and dynamic nature of the immune response and foster training of more robust biomarkers for risk assessment of treatment failure [77,151,152]. As serial sampling of tumor biopsies represents a major hurdle, collection of liquid biopsies is a versatile and non-invasive alternative to predict and evaluate ICI response over time [153,154]. Nowadays, emerging technologies also facilitate the exploration of cancers at the single-cell level. This technology provides a high-resolution insight into the genetic make-up of individual cancer cells as well as the cellular composition of the TIME that cannot be captured by bulk genomics approaches [9,155,156]. In addition to well-established features, including PD-L1 expression, TMB, immune-related gene signatures, MHC and T cell receptor repertoire, clonality of neoantigens and tumor heterogeneity, the balance between immune surveillance and tolerance appears highly context dependent and critically depends on an increasing list of potential variables. These variables include etiological risk factors [157,158], sex differences [159], the microbiome [160], other immune subsets beyond the T cell compartment [161], ternary lymphoid structures [133], the peripheral nervous system [162] and conventional therapeutics [163,164]. These variables should be considered in future explorative and clinical trials addressing the TIME and its alterations during immunotherapy to establish new strategies of combinatorial treatments for HNSCC patients with the final aim to overcome ICI resistance. Finally, integrative analysis of multi-omics data to extract characteristic alterations in cellular and molecular profiles could also contribute to a better understanding of mechanistic principles causing immune-related adverse events (irAEs, Figure 1) or guide treatment-decision making upon progression under immunotherapy [22,165,166].

## Figures and Tables

**Figure 1 cancers-13-01162-f001:**
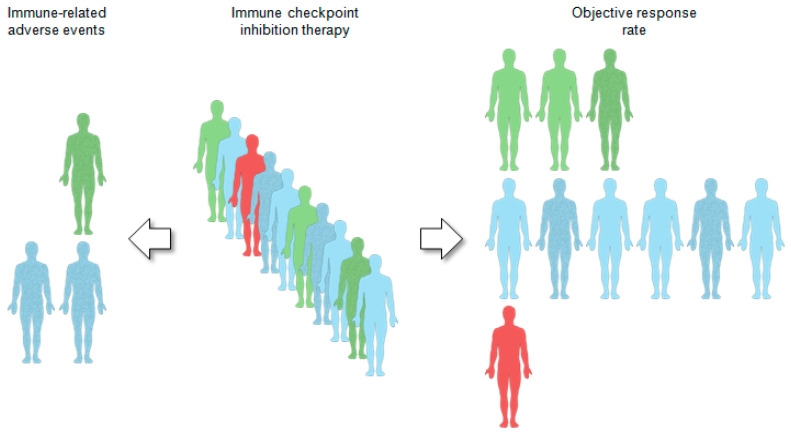
Companion diagnostics for immunotherapy by immune checkpoint inhibition. A limitation of immunotherapy by immune checkpoint inhibition (ICI) is the low response rate and only a small subset of head and neck cancer patients achieves a durable clinical benefit. Main challenge for molecular biomarkers is their predictive value not only to discriminate between responder (green patients) and non-responder (blue patients), but ideally also to assess the risk for a hyper-progressive disease (red patient) under ICI therapy. Establishment of biomarkers that enable early identification of patients at higher risk for severe immune-related adverse events (mottled patients) is another emerging research field of unmet medical need.

**Figure 2 cancers-13-01162-f002:**
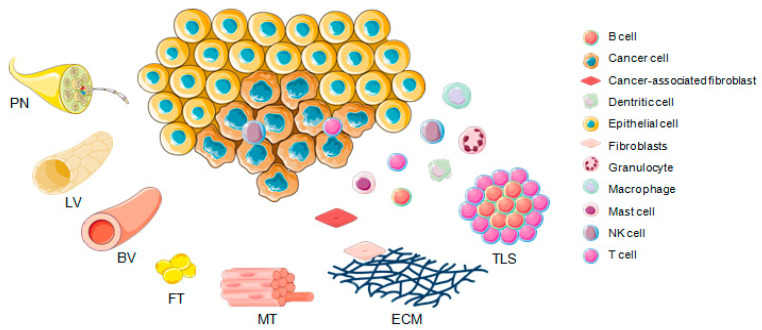
The tumor microenvironment. Structural components of the tumor microenvironment are blood (BV) and lymph vessel (LV), the extracellular matrix (ECM), adipose (AT) and muscle tissue (MT), peripheral nerves (PN), and tertiary lymphoid structures (TLS).

**Figure 3 cancers-13-01162-f003:**
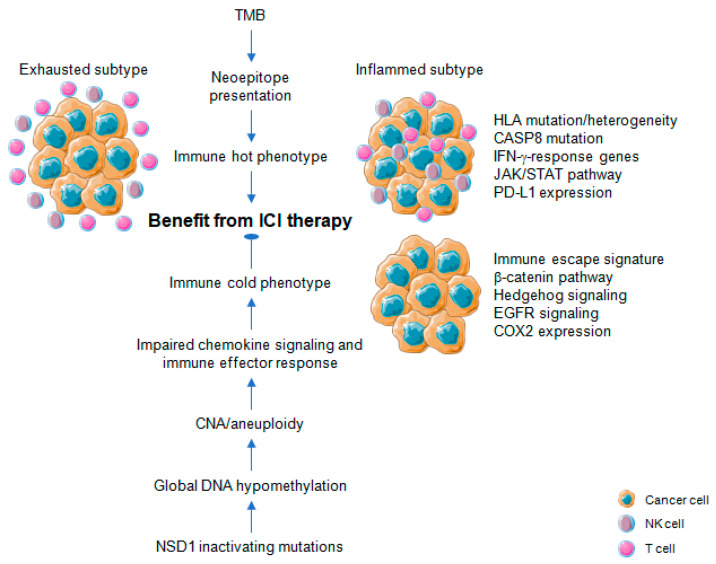
Association of molecular traits related to cancer immune phenotypes and efficacy of immune checkpoint inhibition therapy for head and neck cancer. Tumor mutational burden (TMB) is an indirect measure of tumor antigenicity generated by somatic mutations and neoepitope presentation, causing an immune hot phenotype. The immune hot phenotype emerges as either inflamed or exhausted subtype. An immune cold phenotype is characterized by impaired chemokine and immune effector response, accompanied by specific patterns of copy number alterations (CNA) or aneuploidy. Global DNA hypomethylation promotes chromosomal instability, particularly involving large-scale alterations leading to aneuploidy and has been observed in tumors with NSD1 inactivating mutations. Distinct molecular traits are associated with immune hot or cold phenotypes some of which serve as predictive markers for the efficacy of immune checkpoint inhibition (ICI) therapy.

**Table 1 cancers-13-01162-t001:** Immune-related gene signatures in pan-cancer and HNSCC studies.

Study	Tumor Type	Data Format	Data Source	Signature	Immuno-Phenotype	Mutational Landscape ^1^
Rooney et al. 2015 [83]	pan-cancer	RNA-seq	TCGA	GZMA and PRF1 transcript signature	immune cytolyticactivity	somatic mutations in HLA, B2M, CASP8 copy number changes in CD274, ALOX12B/15B
Keck et al. 2015 [89]	HNSCC	RNA-seq, microarray platforms	TCGA, GSE40774	821-gene signature	immune mesenchymal subtype (IMS)	copy number changes in 3q26 (PIK3CA, SOX2, TP63), 6p21 (VEGFA), 7p11 (EGFR)
Mandal et al. 2016 [54]	HNSCC	RNA-seq	TCGA	ssGSEA scores of tumor-infiltrating immune cell populations and immune signaling molecules	immune-high vs. immune-low	global copy number changes
Ayers et al. 2017 [90]	pan-cancer	NanoStringnCounter platform	KEYNOTE-001, KEYNOTE-012, KEYNOTE-028	T cellinflamed GEP (*n* = 18 genes)	clinical response after pembrolizumab therapy	n.d.
Thorsson et al. 2018 [91]	pan-cancer	RNA-seq	TCGA	Five immune expression signatures (selected out of *n* = 160 signatures)	wound healing, IFNγ dominant, inflammatory, lymphocyte depleted, immune-logically quiet, TGFβ dominant	somatic mutations in CTNNB1, NRAS, IDH1, BRAF, TP53, CASP8
Tamborero et al. 2018 [92]	pan-cancer	RNA-seq	TCGA	GSVA scores of selected immune cell populations (*n* = 16)	six immune-phenotypes with growing abundances of cytotoxic cells	somatic mutations in HLA, B2M, CASP8 copy number changes in PDL1
Jiang et al. 2018 [93]	pan-cancer	RNA-seq, microarray platforms	TCGA, PRECOG, METABRIC	T cell dysfunctional signature	Tumor Immune Dysfunction and Exclusion (TIDE)	n.d.
Chen et al. 2018 [7]	HNSCC	RNA-seq	TCGA	ssGSEA scores of gene expression signatures related to immune pathways	immune class with active or exhausted immune subtypes	global copy numberchanges
Saloura et al. 2019 [66]	HNSCC	RNA-seq, microarray platforms	TCGA, GSE40774	12-chemokine geneexpressionsignature	Tcell-inflamedphenotype	somatic mutations in NSD1, CASP8 copy number changes in EGFR, CD274
Feng et al. 2020 [55]	HNSCC	RNA-seq, microarray platforms	TCGA, GSE40774, GSE117973, GSE39368, GSE65858	CIBERSORT scores of selected immune cell subsets strongly associated with PD-L1 and IFN-γ expression	hot vs. cold immune phenotypes	somatic mutations in CASP8, EP300, TP53 copy number changes in 3p, 5q, 7p, 9p

^1^ selected, GEP = gene expression profile, GSVA = gene set variation analysis, n.d. = not determined, ssGSEA = single sample gene set enrichment analysis.

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
