# Peer review of "Immune-Related Mutational Landscape and Gene Signatures: Prognostic Value and Therapeutic Impact for Head and Neck Cancer"

_cancers, 2021, doi:10.3390/cancers13051162_

Round 1
Reviewer 1 Report
In this review, the authors aims to provide a holistic view on how the immune landscape dictates the tumor fate. And also they reviewed the multi-omics data contribute to the current knowledge on the accuracy of predictive biomarkers and on a broad range of factors influencing the response to immunotherapy in head and neck cancer.
- Most of the current studies on immunotherapy investigating the melanoma, colon cancer or pan-cancer analysis. Although not many studies focused on immunotherapy in head and neck cancers, the authors are encouraged to focus more on head and neck cancers as the title of the article.
1.1. The paragraph starting from line 194 is mostly focusing on melanoma.
1.2. The section “Immune-related gene signatures in pan-cancer studies” discussed mostly on pan-cancer results.
- The typing of the article should be checked carefully and checked, for example:
The “andtobacco… andinfection” typing error in line 32, “andcancer” in line 35 and etc..
- The legend for Figure 3 is overlapping in page 4 and 5. It should be adjusted to put all the legend in one single page.
Author Response
Review #1
In this review, the authors aims to provide a holistic view on how the immune landscape dictates the tumor fate. And also they reviewed the multi-omics data contribute to the current knowledge on the accuracy of predictive biomarkers and on a broad range of factors influencing the response to immunotherapy in head and neck cancer.
- Most of the current studies on immunotherapy investigating the melanoma, colon cancer or pan-cancer analysis. Although not many studies focused on immunotherapy in head and neck cancers, the authors are encouraged to focus more on head and neck cancers as the title of the article.
1.1. The paragraph starting from line 194 is mostly focusing on melanoma.
As requested, we have included additional information on HNSCC at pages 6-7 lines 216-227 and 240-258 of the revised manuscript
1.2. The section “Immune-related gene signatures in pan-cancer studies” discussed mostly on pan-cancer results.
This section aims to summarize global aspects of immune-related gene signatures in pan-cancer studies, including HNSCC, which is followed by a more specialized section (Immune-related gene signatures in HNSCC) with a stronger focus on recent studies based on HNSCC. Relevant aspects for HNSCC were already highlighted in the original manuscript at page 6 lines 234-239 (page 7 lines 281-286 in the revised manuscript), or have been included in the revised manuscript at page 8 lines 299-304 and 306-307.
- The typing of the article should be checked carefully and checked, for example:
The “andtobacco… andinfection” typing error in line 32, “andcancer” in line 35 and etc..
A careful proof-reading was conducted and typing/formatting errors have been corrected in the revised manuscript.
- The legend for Figure 3 is overlapping in page 4 and 5. It should be adjusted to put all the legend in one single page.
Has been done as requested (see page 5 lines 164-174 of the revised manuscript)

Reviewer 2 Report
The authors have described how the immune landscape influences the tumor fate and how integrative analysis of multi-omics data can contribute to improving this. Reviewing the current knowledge on the accuracy of predictive biomarkers and factors influencing the response to immunotherapy in HNSCC is important. Therefore, this is an important report because the response rate to immunotherapy in HNSCC is low and a small subset of patients get benefited from this. However, this could be improved with some additional modifications.
The suggestions are as follows:
1) A separate section/subheading on the immunological landscape of HPV positive and negative HNSCC should be added. This is crucial as the immunological landscape should be different due to the presence of HPV in the cells. Although HPV-related information is included on page 4, 1st paragraph, and page 7, line 289 to 298 but providing additional details in a separate heading would be nice.
2) Please consider summarizing the immune-related gene signatures in pan-cancer studies and HNSCC in the form of a table, highlighting the crucial genes in it.
3) Field cancerization and tumor recurrence is a frequent phenomenon in HNSCC. Compiling details on the immunological landscape of field change cancerization in this review would be good.
4) The “Conclusion and perspectives” section is not clear. It would be good to emphasize and elaborate on the main message/findings of this review here. Certain sentences are complicated to understand in this section, for example, page 8, line 368 -375, page 9, line 357 -362 & page 9, line 375 -379 are complex sentences. Please rephrase to make them simple.
5) The authors may consider numbering the subheadings for better comprehension of the readers.
6) Please correct the adjoined words which are there in almost every paragraph. Usually, these happen during interchanging the text formats.
Example: Introduction; paragraph 1
“andinfection”: page1, line 32
“surveillanceby”: page1, line 36
“HNSCC-sisrather”: page1, line 37-38
Author Response
Review #2
The authors have described how the immune landscape influences the tumor fate and how integrative analysis of multi-omics data can contribute to improving this. Reviewing the current knowledge on the accuracy of predictive biomarkers and factors influencing the response to immunotherapy in HNSCC is important. Therefore, this is an important report because the response rate to immunotherapy in HNSCC is low and a small subset of patients get benefited from this. However, this could be improved with some additional modifications.
The suggestions are as follows:
1) A separate section/subheading on the immunological landscape of HPV positive and negative HNSCC should be added. This is crucial as the immunological landscape should be different due to the presence of HPV in the cells. Although HPV-related information is included on page 4, 1st paragraph, and page 7, line 289 to 298 but providing additional details in a separate heading would be nice.
A separate section on the immune landscape of HPV-positive versus HPV-negative HNSCC has been included at pages 11-12 of the revised manuscript.
2) Please consider summarizing the immune-related gene signatures in pan-cancer studies and HNSCC in the form of a table, highlighting the crucial genes in it.
As requested, a table summarizing pan-cancer and HNSCC studies with immune-related gene signatures was generated and has been included in the revised manuscript as Table 1.
3) Field cancerization and tumor recurrence is a frequent phenomenon in HNSCC. Compiling details on the immunological landscape of field change cancerization in this review would be good.
A new section on the immune landscape and field cancerization has been included at pages 12-13 of the revised manuscript.
4) The “Conclusion and perspectives” section is not clear. It would be good to emphasize and elaborate on the main message/findings of this review here. Certain sentences are complicated to understand in this section, for example, page 8, line 368 -375, page 9, line 357 -362 & page 9, line 375 -379 are complex sentences. Please rephrase to make them simple.
The section Conclusion and perspectives has been revised.
5) The authors may consider numbering the subheadings for better comprehension of the readers.
Has been done as requested.
6) Please correct the adjoined words which are there in almost every paragraph. Usually, these happen during interchanging the text formats.
Example: Introduction; paragraph 1
“andinfection”: page1, line 32
“surveillanceby”: page1, line 36
“HNSCC-sisrather”: page1, line 37-38
A careful proof-reading was conducted and typing/formatting errors have been corrected in the revised manuscript.

Reviewer 3 Report
The aim of the here presented review is to give a current update on the immune – related mutational landscape and gene signature in head and neck carcinoma. The authors rightly postulate that the major therapeutic limitation to ICI therapy in this disease is the low overall response rate, aggravated (1) by the potential risk of hyper – progressive disease and (2) by the high degree of immune-related adverse events. Therefore, an urgent medical demand exists for reliable cellular or molecular biomarkers.
The authors report that only few biomarkers have been established for response evaluation that can be placed in category (1) related to tumor neoepitope burden and category (2) indicative to a T-cell inflamed TME. In their opinion determination of TMB is a well accepted surrogate marker to estimate the tumor neoepitope burden and they postulate TMB as an emerging predictive biomarker for clinical response to ICI therapy.
Indeed, several studies indicated a consistent association between higher TMB and favourable response to ICI therapy, suggesting a potential utility for TMB as a clinical biomarker to guide patient stratification for immunotherapy.
However, the main point of critism is the lacking critical distance to the cited data and the the missing consideration of other publications.
This can be clearly shown considering the TMB topic:
- Until now it is still unclear how metastatic and primary TMB relates back to intratumour heterogeneity, but heterogeneity of a tumour affects on immunotherapy response and polyclonal tumours may bias TMB scores. In this context preiminary data on several tumor entities indicate that TMB classification was inconsistent across multiple regions. As multiregion sequencing is not feasible in clinical practice the utility of TMB as a predictor for immunotherapy response is still a question of debate.
- Pre-clinical data in melanoma and NSCLC cell lines indicate that high-load frameshift indel (insertion and deletion mutations) are highly immunogenic compared with single nucleotide variant load.
- There are also interesting findings which suggeststhat not only the quantity of mutations but also the mutation type is important. In this context, several studies showed that the quality of the mutational burden (i.e., clonal vs. sub-clonal) a had an impact on outcomes, with clonal mutations (homogenous tumours) being associated with better outcomes and a better outcome following ICi therapy compared with sub-clonal (heterogeneous tumours).
- One important aspect which has been largely overlooked to date is the source of tissue sampled for TMB measurement as TMB can be measured from primary or metastatic tumour samples and each of which may cause systematic bias in TMB values. As metastases show higher rates of monoclonal structure due to clonal selection, they present a reduction in overall genetic diversity (‘bottlenecking’). As a systematic evaluation is lacking, very little definitive evidence is available whether metastatic TMB is always higher than primary TMB, but this would support differential thresholds for stratification for immunotherapy. When having a closer look on the published tissue sources used in TMB studies it is obvious that the majority of studies included a mixture of primary and metastatic samples] or did not specify their source of tissue. Taken together it is important to emphasize that the current evidence supporting TMB as a biomarker for immunotherapy response comes from a heterogeneous mix of sample types, including (un)treated primary tissue as well as metastatic tissue.
- In a recently published study on HNSC patients TP53-mutated patients exhibited a higher TMB than TP53 wild type patients, but the expression of most immune checkpoint molecules, such as CD27, CD274, CTLA4, HAVCR2, ICOS, IDO1, LAG3, PDCD1, and TIGIT, was decreased. HNSC patients with TP53-MT exhibited a poor response to ICIs. All of these results indicate the low immunogenicity of TP53-MT patients, which may be one of the mechanisms contributing to their low sensitivity to ICIs.
The figures are not useful to clarify and structure the text
- ICIs can cause severe side effects, therefore the precision of therapy failure is even more important than prediction of therapy response. The requirement of identification of this patient group is addressed in figure 1, but this figure is trivial and the issue is not taken in the text.
- In Figure 2 the multitude of cellular components of the tumor microenvironment are visualized but only few elements are mentioned and discussed in the text.
- Figure 3 is highly undercomplex. To mention just few important findings concerning the influence of immune surveillance. (1) The STING pathway appears to be an important innate sensing pathway for detection of tumours as STING pathway activation within APCs in the tumour microenvironment (TME) drives T-cell priming against tumour associated antigens. (2) ASC, an adaptor molecule, tumor suppressor gene and part of the inflammasome complex influences the immunogenity of a tumor tissue and is frequently inactivated by de novo promoter methylation (in up to 40% of NSCLC patients). (3) In a just published work the authors could show that age > 65 years, TP53-WT, PIK3CA-MT, and ARID1A-MT were associated with prolonged OS in HNSC patients treated with ICIs and that these variables were better outcome predictors than TMB.
A magnitude of reviews with different foci were recently published. The here presented review – at least in its present form – is not intended to offer clarification concerning the complex interaction of immune surveillance and mutational landscape.
Author Response
Review #3
The aim of the here presented review is to give a current update on the immune – related mutational landscape and gene signature in head and neck carcinoma. The authors rightly postulate that the major therapeutic limitation to ICI therapy in this disease is the low overall response rate, aggravated (1) by the potential risk of hyper – progressive disease and (2) by the high degree of immune-related adverse events. Therefore, an urgent medical demand exists for reliable cellular or molecular biomarkers.
The authors report that only few biomarkers have been established for response evaluation that can be placed in category (1) related to tumor neoepitope burden and category (2) indicative to a T-cell inflamed TME. In their opinion determination of TMB is a well accepted surrogate marker to estimate the tumor neoepitope burden and they postulate TMB as an emerging predictive biomarker for clinical response to ICI therapy.
Indeed, several studies indicated a consistent association between higher TMB and favourable response to ICI therapy, suggesting a potential utility for TMB as a clinical biomarker to guide patient stratification for immunotherapy.
However, the main point of critism is the lacking critical distance to the cited data and the the missing consideration of other publications.
This can be clearly shown considering the TMB topic:
- Until now it is still unclear how metastatic and primary TMB relates back to intratumour heterogeneity, but heterogeneity of a tumouraffects on immunotherapy response and polyclonal tumours may bias TMB scores. In this context preiminary data on several tumor entities indicate that TMB classification was inconsistent across multiple regions. As multiregion sequencing is not feasible in clinical practice the utility of TMB as a predictor for immunotherapy response is still a question of debate.
We agree that TMB is at best a limited surrogate biomarker for ICI response and has important limitations as a predictive biomarker, especially when used in isolation. A composite predictor that also includes other critical variables, such as PD-L1 IHC, immune-related gene signatures, MHC and T cell receptor repertoire, clonality of neoantigens and tumor heterogeneity, is urgently needed (see section Conclusion and perspectives at page 13-14 of the revised manuscript). Major challenges for TMB utility and its limitations are summarized and critically discussed in excellent recent studies and review articles (PMID: 30395155, PMID: 31318407, PMID: 32228719, PMID: 33125859, PMID: 33139244), which either have been cited in the original manuscript or are now included in the revised version at page line.
- Pre-clinical data in melanoma and NSCLC cell lines indicate that high-load frameshift indel (insertion and deletion mutations) are highly immunogenic compared with single nucleotide variant load.
We are aware that frameshift insertion/deletions (fs-indels) are a highly immunogenic mutation subtype and beyond pre-clinical data their widespread occurrence and strong immunogenicity has been proven in particular for MSI-H cancers (endometrial, colorectal, and stomach, PMID: 33259803). A more recent study by Ballhausen et al. (PMID: 32958755) demonstrated that frameshift mutation frequency is negatively correlated to the predicted immunogenicity of the resulting peptides, suggesting counterselection of cell clones with highly immunogenic frameshift peptides in MSI colorectal and endometrial cancer. Moreover, fs-indels are degraded through the nonsense-mediated decay (NMD) pathway, though some fs-indels escape degradation, elicit anti-tumor immune responses and are significantly associated with clinical-benefit to ICI therapy in four independent melanoma cohorts (PMID: 32733040). However, the relevance of fs-indels for HNSCC has not been addressed in more detail, so far, and there is an urgent demand for more conclusive pre-clinical and clinical studies.
- There are also interesting findings which suggests that not only the quantity of mutations but also the mutation type is important. In this context, several studies showed that the quality of the mutational burden (i.e., clonal vs. sub-clonal) a had an impact on outcomes, with clonal mutations (homogenous tumours) being associated with better outcomes and a better outcome following ICi therapy compared with sub-clonal (heterogeneous tumours).
This aspect has been included in the revised manuscript at page 4 lines 131-138 and the Section “Conclusion and perspectives” at page 13 lines 519-523.
- One important aspect which has been largely overlooked to date is the source of tissue sampled for TMB measurement as TMB can be measured from primary or metastatic tumour samples and each of which may cause systematic bias in TMB values. As metastases show higher rates of monoclonal structure due to clonal selection, they present a reduction in overall genetic diversity (‘bottlenecking’). As a systematic evaluation is lacking, very little definitive evidence is available whether metastatic TMB is always higher than primary TMB, but this would support differential thresholds for stratification for immunotherapy. When having a closer look on the published tissue sources used in TMB studies it is obvious that the majority of studies included a mixture of primary and metastatic samples] or did not specify their source of tissue. Taken together it is important to emphasize that the current evidence supporting TMB as a biomarker for immunotherapy response comes from a heterogeneous mix of sample types, including (un)treated primary tissue as well as metastatic tissue.
We completely agree and also this aspect is critically discussed in reviews cited in the revised manuscript (see above).
- In a recently published study on HNSC patients TP53-mutated patients exhibited a higher TMB than TP53 wild type patients, but the expression of most immune checkpoint molecules, such as CD27, CD274, CTLA4, HAVCR2, ICOS, IDO1, LAG3, PDCD1, and TIGIT, was decreased. HNSC patients with TP53-MT exhibited a poor response to ICIs. All of these results indicate the low immunogenicity of TP53-MT patients, which may be one of the mechanisms contributing to their low sensitivity to ICIs.
As for TMB and most other variables, TP53-MT should not be considered as isolated feature. As most HPV-positive OPSCC are TP53 wildtype and originate close to lymphatic tissue at tonsils and base of the tongue the association of TP53-MT with differences in both immune checkpoint protein expression and low immunogenicity might be biased by the HPV status. This assumption is supported by our study showing an association between TP53 mutations and a cold immunophenotype for HNSCC, but not other cancer types (PMID: 32161122). It is worth noting that the study be Zhang et al. (PMID: 33363171), which demonstrated lower transcript levels of most immune checkpoint molecules for TP53-MT tumors of the TCGA-HNSC cohort did not adjust their data for HPV status as a strong confounder and association with survival upon ICI therapy was assessed for an independent cohort (MSKCC-HNSC) without providing data on immune checkpoint molecule expression or HPV status. Finally, an emerging body of experimental evidence indicates that not all TP53 mutations are equal and associations with either immune checkpoint protein expression or response to ICI therapy is largely context dependent (e.g. PMID: 32927274, PMID: 32196516). To the best of our knowledge, the link between TP53-MT and response to ICI therapy is not supported by a clinical trial including HNSCC patients.
The figures are not useful to clarify and structure the text
- ICIs can cause severe side effects, therefore the precision of therapy failure is even more important than prediction of therapy response. The requirement of identification of this patient group is addressed in figure 1, but this figure is trivial and the issue is not taken in the text.
We disagree with the statement that figure 1 is trivial and its issue has been discussed several times in the main text (page 2 lines 69-74, page 13 lines 504-506 and page 14 lines 529-533).
- In Figure 2 the multitude of cellular components of the tumor microenvironment are visualized but only few elements are mentioned and discussed in the text.
Figure 2 aims to illustrate the complexity of the tumor microenvironment, including several cellular and structural components. In the main text, we focused on those components for which a causal link with the establishment or maintenance of the immunophenotype is proven by a sufficient amount of experimental and/or clinical evidence. The impact of other components (e.g. TLS and peripheral nerves) requires future studies as mentioned in the last section “Conclusion and perspectives” at pages 13-14 of the revised manuscript.
- Figure 3 is highly undercomplex. To mention just few important findings concerning the influence of immune surveillance. (1) The STING pathway appears to be an important innate sensing pathway for detection of tumours as STING pathway activation within APCs in the tumour microenvironment (TME) drives T-cell priming against tumour associated antigens. (2) ASC, an adaptor molecule, tumor suppressor gene and part of the inflammasome complex influences the immunogenity of a tumor tissue and is frequently inactivated by de novo promoter methylation (in up to 40% of NSCLC patients). (3) In a just published work the authors could show that age > 65 years, TP53-WT, PIK3CA-MT, and ARID1A-MT were associated with prolonged OS in HNSC patients treated with ICIs and that these variables were better outcome predictors than TMB.
Figure 3 summarizes main topics, which have been discussed in the main text and are supported by several independent studies to modulate the efficacy of ICI therapy of HNSCC patients. An illustration of all potential aspects, which might be relevant including those that are not discussed in this review (e.g. STING, ASC, etc.) would be confusing and in our opinion is most likely not constructive.

Round 2
Reviewer 1 Report
The article was well revised.
Author Response
Reviewer #1
The article was well revised.
Reviewer 2 Report
I do not have any concerns.
However, there are still some conjoined words which should be corrected in the final version.
Author Response
Reviewer #2
I do not have any concerns.
However, there are still some conjoined words which should be corrected in the final version.
We apologize for any inconvenience and will ensure the elimination of all conjoined words, which occur while uploading the document to the online system, during final proof-reading of the manuscript.
Reviewer 3 Report
The review in its present form largely benefited from the reviewer comments.
In detail, (1) the „Simple Summary“ now represents the following text, (2) the impact of TMB measurement is now supplemented by an explanatory text, (3) all – cancer findings and HNSCC specific findings are more separated - however only in the headings - and (4) the role of transposable elements and methyltransferases and demethylases in epigenetic changes and immune escape is now added.
Also, the molecular immune subtypes are now discussed in the chapter of current findings concerning the immune landscape of HNSCC. A new important focus is placed on the involvement of computational tools to evaluate high throughput data. The new Table 1 greatly facilitates the understanding of current data concerning immune related gene signatures.
Chapter 5 is completely new written and deals with a comparison of HPV positive and negative tumor characterisitics.
In addition, also chapter 6 is new and incorporates the most important subject of preneoplasia.
*Major points*: However, although the adding of the above mentioned topics clearly improves the manuscript the „old“ and „new“ parts are almost incoherent. Concerning the impact of HPV infection on the biology and outcome of HNSCC patients the statements of the manuscript are even contradictory. In addition, the comments concerning the topic of preneoplasia are not satisfactory as they do not reflect the spatial separation of atypical epithelial cells and subepithelial stroma by the basement membrane. Furthermore, a critical statement on the pro and cons of TCGA data derived bioinformatic analysis is only done in a very short manner
*Minor point*: As already mentioned previously the graphics not really assist readers in understanding the complex theme of immune-related mutational landscape and gene signatures. Taken together, the manuscript improved a lot but is still not able to be published in its present form.
Author Response
Reviewer #3
The review in its present form largely benefited from the reviewer comments.
In detail, (1) the „Simple Summary“ now represents the following text, (2) the impact of TMB measurement is now supplemented by an explanatory text, (3) all – cancer findings and HNSCC specific findings are more separated - however only in the headings - and (4) the role of transposable elements and methyltransferases and demethylases in epigenetic changes and immune escape is now added.
Also, the molecular immune subtypes are now discussed in the chapter of current findings concerning the immune landscape of HNSCC. A new important focus is placed on the involvement of computational tools to evaluate high throughput data. The new Table 1 greatly facilitates the understanding of current data concerning immune related gene signatures.
Chapter 5 is completely new written and deals with a comparison of HPV positive and negative tumor characterisitics.
In addition, also chapter 6 is new and incorporates the most important subject of preneoplasia.
Major points:
However, although the adding of the above mentioned topics clearly improves the manuscript the „old“ and „new“ parts are almost incoherent. Concerning the impact of HPV infection on the biology and outcome of HNSCC patients the statements of the manuscript are even contradictory.
Unfortunately, a more detailed description for this concern is not given andthe reviewer did not provide any specific example for a contradictory statement in our manuscript. Hence, we checked all HPV-related statements, have included a comment at page 4 line 130 and have added more information from clinical trials in Section 5 (page 11 lines 425-442) of the revised manuscript.
In addition, the comments concerning the topic of preneoplasia are not satisfactory as they do not reflect the spatial separation of atypical epithelial cells and subepithelial stroma by the basement membrane.
As requested, we have included the information that a characteristic feature of preneoplasia in the oral cavity is the spatial separation of atypical epithelial cells and subepithelial stroma by the basement membrane at page 12 lines 490-496 of the re-revised manuscript.
Furthermore, a critical statement on the pro and cons of TCGA data derived bioinformatic analysis is only done in a very short manner
We agree that numerous pro and cons exist for utilizing big data, including those from TCGA. However, a critical discussion of promises and pitfalls using big data for basic science and clinical research is multilayered, should not be limited to TCGA data and is far beyond the main scope of this manuscript.
Minor point:
As already mentioned previously the graphics not really assist readers in understanding the complex theme of immune-related mutational landscape and gene signatures.
As already answered in the point-by-point letter for the revised manuscript, we disagree with this statement and as both other reviewers did not raise any concern on the content or quality of the figures, we decided not to change them for the re-revised manuscript.